# *Clostridium perfringens* Type D Epsilon Toxin Causes Blood–Retinal Barrier Microvascular Damage and Diffuse Retinal Vasogenic Oedema

**DOI:** 10.3390/vetsci11010002

**Published:** 2023-12-19

**Authors:** John W. Finnie

**Affiliations:** Division of Research and Innovation, School of Medicine, University of Adelaide, Adelaide, SA 5005, Australia; john.finnie@adelaide.edu.au

**Keywords:** blood–retinal barrier breakdown, *Clostridium perfringens* type D epsilon toxin, rats, retinal vasogenic oedema

## Abstract

**Simple Summary:**

The bacterium *Clostridium perfringens* type D produces a potent epsilon toxin (ETX), which causes severe, and frequently fatal, neurologic disease in sheep and goats. The brain microvasculature (blood–brain barrier or BBB) is the principal target of EXT, which results in abundant vascular fluid and plasma protein leakage. The accumulation of this excess fluid (termed oedema) in the brain leads to a marked increase in intracranial pressure and neurologic disturbance. Although the BBB resembles that of the retina (blood–retinal barrier or BRB) in structure and function, the effect of ETX on the ruminant retina has not been studied. However, in rats given ETX, similar microvascular injury to that described in sheep and rat brains was found in the retina, leading to generalized vasogenic retinal oedema. Increased retinal vascular permeability was demonstrated immunohistochemically by leakage of the osmotically active plasma protein, albumin, and microvascular injury was confirmed by loss of the marker of an intact BRB in this species, endothelial barrier antigen. These ocular findings in rats make it likely that similar retinal pathology will be found in ETX-intoxicated sheep and this rodent model could be useful to test potential drug treatments for retinal oedema in humans and domestic animals.

**Abstract:**

*Clostridium perfringens* type D epsilon toxin (ETX) causes severe retinal microvascular endothelial injury in the rat. The resulting blood–retinal barrier (BRB) breakdown leads to increased vascular permeability, which was detected immunohistochemically by the extravasation of plasma albumin as a vascular tracer, and ensuing severe, diffuse, vasogenic retinal oedema. This microvascular damage was also confirmed by a loss of endothelial barrier antigen, a marker of an intact BRB in rats. Since similar microvascular lesions are found in EXT-exposed laboratory rodent and sheep brains, and the BRB resembles the BBB, they are also likely to occur in the eyes of naturally epsilon-intoxicated sheep and goats, but this remains to be determined. Moreover, while retinal oedema is a common and important component of many human and veterinary ocular disorders, more effective treatments are required. Accordingly, the retinal vasogenic oedema reliably and reproducibly induced by ETX in rats provides a useful model in which to study the pathogenesis of retinal oedema development and evaluate its prevention or amelioration by putative pharmacological interventions.

## 1. Introduction

The potent ETX of *Clostridium perfringens* type D principally targets the brain, with a clinical neurologic presentation, but also produces lesions in the heart, lungs, and kidney [1,2,3]. However, while this toxin accumulates in the eye [4], and blindness commonly occurs in ETX-intoxicated sheep [5], ocular pathology has only been studied to date in the rat. 

The major focus of this review is the ETX-induced breakdown of the blood–retinal barrier (BRB) in a rat model, with resulting increased vascular permeability and diffuse, vasogenic retinal oedema. Firstly, however, the nature of ETX and the naturally occurring disease it produces in ruminant livestock is outlined. Since the principal ocular target of ETX is the retinal microvasculature, the structure and function of the BRB, which maintains homeostasis of the retinal microenvironment, is described. The consequence of BRB disruption is retinal oedema and, accordingly, the formation and clearance of this oedema will then be discussed as a prelude to describing ETX ophthalmotoxicity in the rat. Finally, the implications of ETX oculotoxicity in the rat for an improved understanding of ETX intoxication in ruminants will be discussed, together with the potential usefulness of this rodent model for testing improved therapies for retinal oedema in humans and animals. 

## 2. *Clostridium perfringens* Type D Epsilon Toxin

The anaerobic, spore-forming bacillus, *Clostridium perfringens*, produces about 17 different toxins, which enables this bacterium to cause disease by a range of pathogenetic mechanisms [6]. The bacterium is classified A to G [7], which is a toxin-based typing classification based on elaboration of one or more of six exotoxins, namely alpha, beta, epsilon, iota, enterotoxin and necrotic enteritis B-like (Net B) toxins (Table 1).

Type D strains produce two of these toxins, alpha and ETX, but ETX is the main virulence factor of this toxinotype. ETX is secreted as a relatively inactive, ~33 kDa polypeptide, termed epsilon prototoxin. Intestinal enzymes in ruminants, or other proteases produced by *Clostridium perfringens*, proteolytically activate the prototoxin to the active ~29 kDa ETX by removing the C-terminal amino acids [8,9]. Of the clostridial toxins, only botulinum and tetanus toxins are more potent than ETX [1].

The naturally occurring neurologic disease produced by ETX in ruminant livestock, particularly sheep and goats, is severe and frequently fatal. Small numbers of *Clostridium perfringens* type D are sometimes found in the small intestine, but the quantity of ETX produced is relatively innocuous. However, when the microbial balance in the gut is disrupted, especially following a starch overload from ingested lush pasture or grain, there is rapid proliferation of this clostridial bacterium and large amounts of ETX can be elaborated. ETX then facilitates its own absorption into the systemic circulation by increasing intestinal permeability, thereby initiating a purely enterotoxemic phase. When high levels of ETX are absorbed from the intestine into the systemic circulation, this neurotoxin binds to microvascular endothelial receptors in the brain. As an aerolysin family, pore-forming toxin, ETX first binds to a receptor, then uses lipid rafts and caveolins to oligomerize into a heptameric pre-pore on the cell surface. The toxin extends a β-hairpin loop into the membrane lipid bilayer to form an active pore, which leads to a rapid decrease in cytoplasmic K^+^ levels, an influx of Na^+^ and Cl^−^ into the susceptible endothelial cell, and eventually cell death [10]. 

The vascular damage produced by ETX causes increased permeability of cerebral capillaries and venules and extravasation of abundant fluid and plasma protein into the brain parenchyma. This vasogenic oedema becomes diffusely distributed throughout the brain, causing a marked rise in intracranial pressure and severe neurologic disturbance, coma, sometimes cerebellar herniation and, eventually, often death [1,2,3].

ETX also binds to, and damages, the microvascular endothelium in the lungs, heart, and kidneys. After the intravenous injection of ETX in mice, large quantities of toxin accumulate in the kidneys, in addition to the brain, and it was posited that this renal sequestration of ETX could be neuroprotective by limiting the quantity of toxin available to reach the brain [4]. 

## 3. The Blood–Retinal Barrier

Since ETX-induced retinal injury in the rat is fundamentally vasculocentric [1,2,3], with severe damage sustained by the microvascular endothelium, it is pertinent to briefly outline the structure and function of the protective, regulatory barrier between circulating blood and the retinal extracellular space. 

The retina is highly metabolically active, has the greatest oxygen consumption of any tissue in the body, and consumes 8% of the basal metabolic rate [11,12,13]. Accordingly, the retinal microenvironment, particularly that of neurons and photoreceptors, must be tightly regulated to maintain optimal homeostasis and protect the delicate neural tissues from fluid overload and osmotic stress. This function is performed, like the blood–brain barrier (BBB), by the BRB [11,12,13]. 

There are two principal barriers between circulating blood and the eyes, namely the blood-aqueous barrier and the BRB. The former is comprised of a ciliary body and iris epithelial cells, and the iridial microvascular endothelium, both of these barriers possessing “leaky-type” intercellular tight junctions. The BRB consists of an inner (iBRB) component of retinal microvascular endothelial cells and outer (oBRB) retinal pigment epithelial (RPE) cells. These BRB cells are interconnected by tight junctions (*zonulae occludentes*) which, in addition to the plasma membrane of barrier-forming cells, are multiprotein junctional complexes forming the principal intercellular (paracellular) barrier to leakage of solutes and water. This barrier is not absolute, but selectively permeable [12,14]. Retinal capillary endothelial cells have more intercellular tight junctions than other tissues and the smallest intercellular space, thereby effectively preventing free leakage of fluid and protein. Retinal interstitial spaces, therefore, normally contain scant water, and this relatively dehydrated state creates tissue transparency and facilitates optimal light transmission [13,14,15]. 

Since the retina is the most metabolically active tissue in the body, it has a dual blood supply. The photoreceptors in the outer retina are supplied by the choroid, and the inner by the ciliary, arteries, the exception to the latter being primates, which have a single central retinal artery [11,13]. In rats, there are six main arterial and venous branches, which run from the optic disk towards the periphery of the retina. Retinal capillaries are tortuous and cylindrically arranged in two layers: a superficial network derived from arterioles and a deeper, more densely arranged network connected to venules [16]. Moreover, in most species, including dog, cat, cow, sheep, rat, and mouse, the majority of the inner retinal surface is traversed by blood vessels (holangiotic) [17]. The retinal microvascular endothelial cells forming the iBRB are surrounded by a basement membrane, and modulating and stabilizing pericytes and glial (astrocytes, Muller cells, and microglia) cells. These cellular elements, in concert with neurons, are functionally interdependent and comprise the neurovascular unit. The RPE rests on Bruch’s membrane, separating the fenestrated choriocapillaris from the retina, thus regulating the access of nutrients from the systemic blood to the photoreceptors and protecting the outer retina from an influx of fluid [12]. 

BRB breakdown plays an important role in the pathophysiology of the most frequent and clinically significant human blinding diseases, namely diabetic retinopathy (DR) and age-related macular degeneration (AMD). The development of DR and AMD is directly attributable to BRB disruption, which involves the iBRB and oBRB, respectively. Incompetence of the BRB allows access of a protein-rich fluid to the retinal parenchyma, resulting in macular oedema, the major cause of vision impairment in these ocular diseases [12,13]. Retinal oedema is also an important component of some veterinary ophthalmic disorders [17]. 

Blood-ocular barriers sequestrate the retina from the immune system and create an environment of immune privilege, abetted by the immunosuppressive actions of BRB and RPE cells [12].

## 4. Mechanisms of Retinal Oedema Formation and Clearance

Under normal physiologic conditions, the retina maintains a low level of hydration, which is necessary for optimal visual acuity. However, when the BRB is disrupted by ETX, the resulting increased vascular permeability permits the leakage of fluid and plasma proteins, particularly osmotically active albumin, into the retinal interstitial space. The retina then attempts to resolve this excess fluid (oedema) by several mechanisms [15].

Intraocular pressure normally causes a slow, continuous leakage of water into the extracellular space of the retina, which is then moved by active transport across the RPE, finally being attracted into the choroidal vasculature by its high osmotic pressure. This protein gradient from the retina to the choroid causes a permanent outflow of water and, as a result, the water content of the retina is usually minimal. The retina, especially the centrally located macula responsible for high visual acuity, is functionally very sensitive to any change in its state of hydration, which must be low to allow the optimal transmission of light to the photoreceptors [18,19].

Both active and passive forces are required to efficiently move water across, and out of, the retina. Passive forces, which are strong, are the intraocular pressure pushing water into, and high osmotic pressure within the choroid drawing water out of, the retina. These passive forces are abetted by active transport across the RPE towards the choroid [20].

Water and solutes move freely across the inner limiting membrane (ILM) of the retina (Table 2), which is composed of collagen and matrix material, but the retinal interstitial space is long and convoluted and provides considerable resistance to water movement. They eventually reach the partially restrictive external limiting membrane (ELM), which is composed of Muller cells and photoreceptors connected by adherens junctions (*zonulae adherentes*). These junctions are less effective than tight junctions of the BRB and RPE and permit egress of large proteins such as albumin, the major regulator of oncotic pressure. Nevertheless, large molecules do not readily traverse the retina and their removal is inversely proportional to their size. They tend to pool behind the ELM and osmotically retain water, remaining in this location for a considerable duration of time, before some eventually diffuse into the vitreous cavity or subretinal space [18,19,20].

When no solutes are present, excess water is efficiently removed by active transport across the RPE, which fulfills this important role in the absence of retinal lymphatics. While the RPE has a large capacity for pumping water, flow resistance within the retina limits the amount of water which can be presented to the absorptive surface of the RPE. When the capacity of RPE active transport is exceeded, water is retained, and retinal oedema develops. Water eventually reaches the choroid, its fenestrated capillaries permitting the passage of even large molecules such as albumin [18].

In addition to the RPE, fluid resorption from the retina is performed by Muller cells, the principal glial cells of the retina, and astrocytes. The processes of these glial cells closely invest retinal capillaries and provide a functional link between blood vessels and neurons. Water fluxes through these glia, and the RPE, are facilitated by specialized water transporting proteins, the aquaporins, with RPE cells expressing aquaporin-1 and Muller cells aquaporin-4. Muller cells dehydrate the retinal parenchyma, while the RPE removes water from the subretinal space [21].

The transport of molecules to and from the retina is tightly controlled by capillary endothelial and RPE cells, with both transcellular and paracellular routes being operative. Tight junctions are the main regulators of paracellular transport between endothelial and epithelial cells at the BRB and RPE, respectively. The breakdown of tight junction proteins, increased transendothelial transport, and damage to, and/or loss of, constituent BRB cells (endothelial, pericytic, and glial) disrupt the BRB and result in increased vascular permeability. This leads to an imbalance between fluid entry and exit in the retina and the accumulation of water (vasogenic oedema) and plasma proteins in the extracellular space, thereby increasing retinal thickness and impairing vision [18,19].

The normal osmotic pressure of the retina is almost zero but, with increased BRB permeability, proteins tend to aggregate behind the ELM, thus elevating oncotic pressure and promoting further oedema formation. The increased water content of the retina can be intracellular in neurons and glia, especially Muller cells and astrocytes (cytotoxic oedema), or extracellular, the latter being more frequent and clinically significant. However, oedema which is initially vasogenic eventually becomes a mix of vasogenic (interstitial) and cytotoxic (cellular) water accumulation. Retinal oedema contributes to compression of neurons, nerve fibers and blood vessels, and elongation of diffusion routes for metabolic substrates and oxygen [18,21].

Retinal oedema is a hallmark feature of many human and veterinary ocular diseases and, in most, there is some degree of damage to both the BRB and RPE. However, in some disorders, injury to one barrier type predominates [15]. Macular oedema, which is defined as fluid accumulation in the outer plexiform and inner nuclear layers (Table 2) of the retina, and swollen Muller cells, is the final common pathway of many ocular and systemic diseases and responsible for loss of visual acuity. It is usually slowly progressive, but can have a sudden onset. This central retinal area is predisposed to oedema development because of its high metabolic rate, the loose structure of the outer plexiform layer, and inability of the macula to resorb water into the vasculature [19,22].

## 5. Clostridial Epsilon Ophthalmotoxicity in the Rat

Although it is well-established that the prime target of ETX is the cerebral microvasculature [1,2,3], with BBB disruption leading to severe, diffuse, vasogenic oedema, this potent toxin also accumulates in the eye [4]. Since the BRB resembles the BBB in many important respects [12,13], studies were conducted in the rat to determine whether ETX injured the retinal microvasculature in a similar manner to that of cerebral blood vessels, and produced vasogenic oedema.

In the first study of ETX-induced retinal injury in rats [23], six-week-old, Sprague Dawley animals were given an intraperitoneal injection of a 1:300 dilution of a batch of ETX, which was equivalent to 69,000 mouse minimum lethal doses of this toxin. At 3 h post-injection of toxin, rats were killed by perfusion fixation with 4% paraformaldehyde, with one eye then fixed in Davidson’s fixative for light microscopy and the other with 2.5% glutaraldehyde for electron microscopy. For the immunohistochemical detection of plasma albumin extravasation, a goat anti-rat albumin antibody was used. Control rats were given a similar volume of physiological saline and their eyes were similarly processed. When rats were given this acute dose of ETX [23], severe ultrastructural retinal microvascular injury was found. The most severe endothelial damage was characterized by marked attenuation and increased electron density (Figure 1). Prior to the development of endothelial coagulation necrosis, there was frequent blebbing of the luminal surface and occasional platelet adhesion (Figure 1), but not thrombosis. These retinal microvascular lesions resembled those found in the brain microvasculature of sheep [2,3] and mice [24] exposed to ETX (Figure 2). To further examine this BRB breakdown, immunohistochemical studies used endogenous albumin extravasation as a surrogate marker of increased vascular permeability [23]. In ETX-treated rats, there was diffuse albumin extravasation in all retinal layers, which was particularly prominent around small blood vessels, and albumin also flooded the cytoplasm of damaged endothelial cells (Figure 3). This widespread retinal albumin leakage confirmed the presence of generalized vasogenic oedema and resembled the albumin extravasation found in ETX-exposed sheep (Figure 4) [2,3] and rat brains [23]. Microvascular albumin leakage is particularly deleterious to the retina as regulation of albumin transport is critical for maintaining appropriate protein gradients in the retina and subsequent fluid movement [12,13].

The severe microvascular damage found in the brain, and retina, of rats after acute ETX exposure is supported by the finding that this toxin produces a rapid and dose-dependent cytotoxic effect on cerebral microvascular endothelial cells in vitro [25]. ETX binding to brain endothelial cells appears to be mediated by the myelin and lymphocyte protein (MAL) receptor [26]. 

The retinal [27], and brain [28], vasogenic oedema found in rats given ETX, was correlated with loss of capillary endothelial barrier antigen (EBA), which is a sensitive and specific marker of an intact and functionally competent BBB and BRB in this species [29,30,31,32]. Although EBA is strongly expressed by brain and retinal capillaries possessing a BBB or BRB, respectively, it is absent or only weakly expressed by fenestrated, non-BBB and BRB blood vessels [29,32]. The rat-specific BBB and BRB marker protein, EBA, is a membrane protein triplet of 23.5, 25 and 30 kDa, which is uniformly expressed on the luminal aspect of brain and retinal microvessels [31]. 

A second study designed to detect EBA changes in ETX-treated retinas used a similar experimental protocol to that examining retinal morphological changes [23], with EBA immunohistochemically labelled by a mouse monoclonal anti-EBA antibody [27]. In ETX-exposed rat brains (Figure 5) [28] and eyes (Figure 6) [27], there was heterogeneous loss of microvascular EBA immunoexpression, which was either complete or partial. Less commonly, there was no apparent loss of EBA immunopositivity. Loss of EBA immunoreactivity in eyes and brain also correlated with marked extravasation of plasma albumin as a vascular tracer and marker of increased vascular permeability. By contrast, the microvascular endothelium in control rat brains (Figure 5) and retinas (Figure 6) was uniformly immunolabeled with EBA [27,28].

As in rat brains exposed to ETX [28], the heterogeneity of EBA loss in retinal microvessels [27] suggested that these blood vessels were not equally susceptible to EXT-induced endotheliotoxicity, perhaps because only a subset of retinal microvessels possess receptors for ETX. This apparently selective BRB breakdown may also be explained by the fact that, in the similar BBB, there is a heterogeneous population of endothelial cells, reflecting the functional diversity of different brain regions [33]. This endothelial heterogeneity is also found within individual brain microvessels [34] and likely also pertains to the BRB endothelium. Moreover, in experimentally intoxicated mice, ETX causes brain damage in a dose- and time-dependent manner [2,3], implying that only microvessels exposed to sufficient ETX sustain damage. 

EBA is believed to be an important component of tight junctions in rat brain and retinal microvessels, and its expression is correlated with a functionally intact BBB and BRB. It is unclear whether loss of EBA plays a causative role in opening the BBB and BRB, or is a consequence of altered barrier integrity. Administration of an anti-EBA antibody to rats resulted in opening of the BBB and extravasation of a vascular tracer [34,35,36,37], suggesting that the former mechanism is likely to be operative. This anti-EBA antibody also showed that opening of the BBB was monophasic and transient and, once barrier function was restored, it maintained its integrity [35]. However, another study [38] showed that opening of the injured BBB preceded a reduction in endothelial EBA immunopositivity.

EBA is found in an endothelial cytoplasmic pool, from which it may replenish luminal membrane EBA. ETX diminishes endothelial cytoplasmic immunolabeling of EBA before reducing luminal immunoreactivity, suggesting that it could impede EBA synthesis and progressively deplete endothelial EBA at the luminal surface. Moreover, EBA expression appears to be highly regulated and may alter temporally, depending on the prevailing microenvironmental milieu [34,35,36,37,39].

## 6. Future Studies

Although ETX-induced brain and cardiorespiratory lesions, including microvascular damage, have been well described in the spontaneously arising disease caused by this toxin in sheep and goats [1,2,3], ocular pathology in these species has not been investigated. Blindness is a common clinical finding in EXT-intoxicated sheep [5], but whether this is caused by brain or ocular pathology, or both, remains to be determined. If ocular lesions are found in sheep, they could be diagnostically useful, although retinal oedema can sometimes be difficult to appreciate in routinely immersion-fixed and stained eyes. However, extravasated plasma albumin could be detected immunohistochemically in formalin-fixed, paraffin-embedded sheep retinas, as has been performed on ETX-exposed brains in this species [3], although perfusion-fixed tissues are preferable to optimally label albumin with this technique. 

When sheep brains are exposed to large doses of ETX, an acute, and often rapidly fatal, neurologic syndrome develops. However, when toxin exposure is smaller, or these animals are partially immune to ETX, a more protracted clinical course ensues, and sometimes bilaterally symmetrical necrotic foci are found in certain selectively vulnerable brain regions (termed focal symmetrical encephalomalacia) [2,3]. Similarly, mice given multiple, subacute doses of ETX develop multifocal necrotic areas in the brain [24]. It remains to be determined whether rats given subacute, and/or repeated, doses of ETX similarly develop multifocal cerebral necrosis after a more prolonged period of intoxication. Moreover, under this dosage regimen, retinal necrosis may also be found, rather than the more fulminant vasogenic oedema produced by acute ocular exposure to ETX. 

Since treatment options for retinal oedema, which can markedly impede visual acuity, are currently limited in humans and domestic animals, this rat model of ETX-induced vasogenic retinal oedema could be useful to test potential therapeutic interventions designed to prevent or ameliorate this excessive water accumulation.

## 7. Conclusions

The present review has highlighted that, in addition to disrupting the BBB and causing severe, diffuse cerebral oedema in ruminant and laboratory animal species, ETX can similarly injure the BRB microvasculature, which structurally and functionally resembles the BBB, and produce marked retina oedema in the rat. ETX-induced BRB breakdown was confirmed by the immunohistochemical detection of extravasated plasma albumin, and loss of endothelial EBA immunoreactivity, the latter a marker of an intact BRB in this species. BRB disruption then led to increased vascular permeability and leakage of plasma proteins, notably albumin, and water, this diffuse retinal vasogenic oedema probably overwhelming the clearance capacity of the RPE, and Muller and astrocytic glial cells. While ETX has been shown to produce vasculocentric retinal injury, and ensuing oedema, in rats, it remains to be determined whether similar ocular damage occurs in ETX-intoxicated ruminants, especially sheep and goats. 

## Figures and Tables

**Figure 1 vetsci-11-00002-f001:**
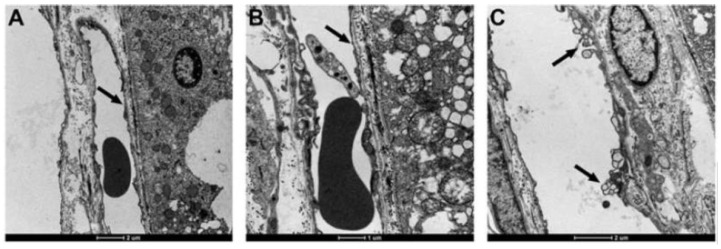
Electron micrograph of ETX-induced coagulation necrosis of the retinal microvasculature (arrows) in a rat, this damage evidenced by marked attenuation and electron-density of the endothelium (**A**,**B**). Less severe injury is represented by focal blebbing and multivacuolation (**C**). Uranyl acetate and lead citrate. Bar = 2 µm.

**Figure 2 vetsci-11-00002-f002:**
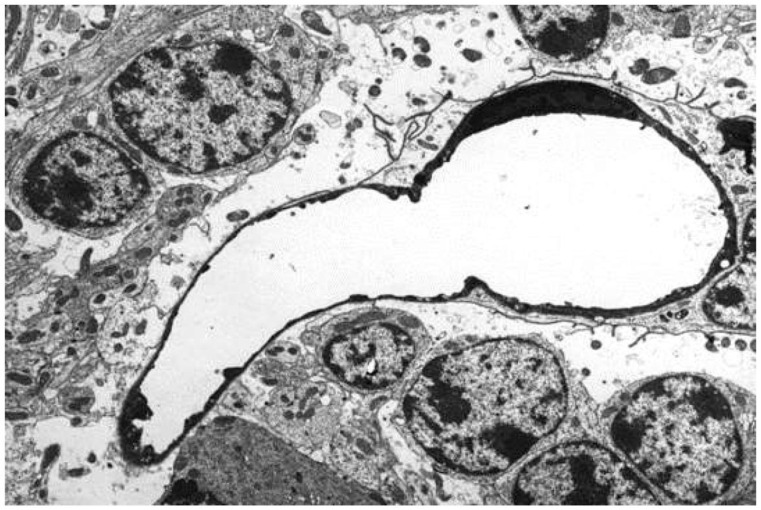
Electron micrograph of similar capillary damage to Figure 1 produced by ETX in the cerebellar granular layer of a sheep. As in rats, the endothelium is markedly attenuated and electron-dense, with nuclear pyknosis. Perivascular astrocytic end-foot processes are severely swollen as a result of increased vascular permeability. Uranyl acetate and lead citrate. ×3750.

**Figure 3 vetsci-11-00002-f003:**
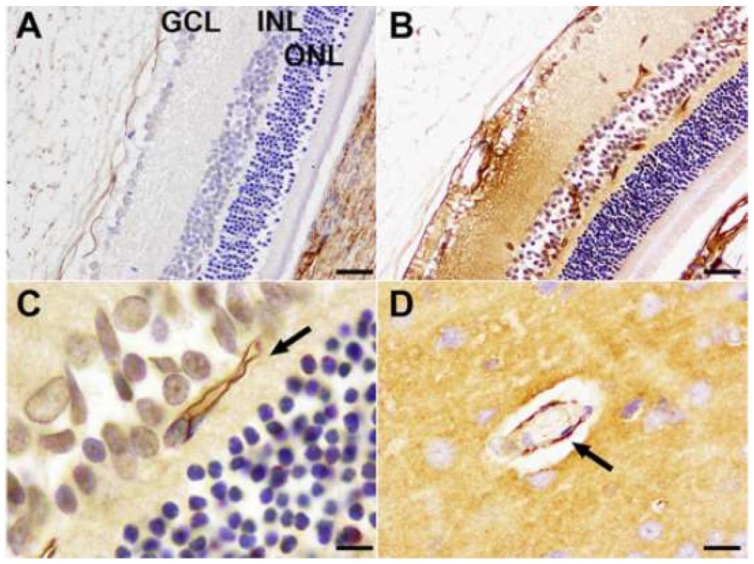
In a control, ETX-untreated rat retina (**A**), all layers are free of albumin immunostaining (GLC = ganglion cell layer; INL = inner nuclear layer; ONL = outer nuclear layer). By contrast, in an ETX-treated eye (**B**), microvascular injury has led to diffuse plasma albumin leakage and immunostaining of the retina. A microvessel (**C**) is prominently immunolabeled for albumin due to flooding of the damaged endothelium (arrow). ETX-treated retinal lesions resemble those found in an ETX-exposed rat brain (**D**), with marked albumin immunostaining of the injured endothelium (arrow), The surrounding brain parenchyma shows diffusely albumin immunopositivity due to robust vascular leakage of this plasma protein. Albumin immunohistochemistry. Bar = 15 µm.

**Figure 4 vetsci-11-00002-f004:**
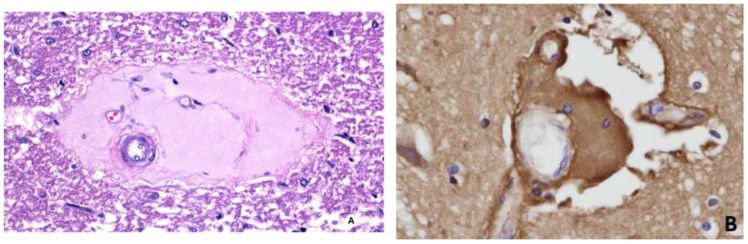
Similar vascular changes to those shown in Figure 3 in the rat retina were found in a routinely fixed and stained, acutely ETX-intoxicated, sheep brain, with characteristic perivascular deposition of extravasated plasma protein-rich fluid (**A**). This leaked proteinaceous fluid corresponded, in a similarly injured microvessel (**B**), to immunohistochemically labelled albumin extravasation. H&E (**A**); albumin immunohistochemistry (**B**) ×200.

**Figure 5 vetsci-11-00002-f005:**
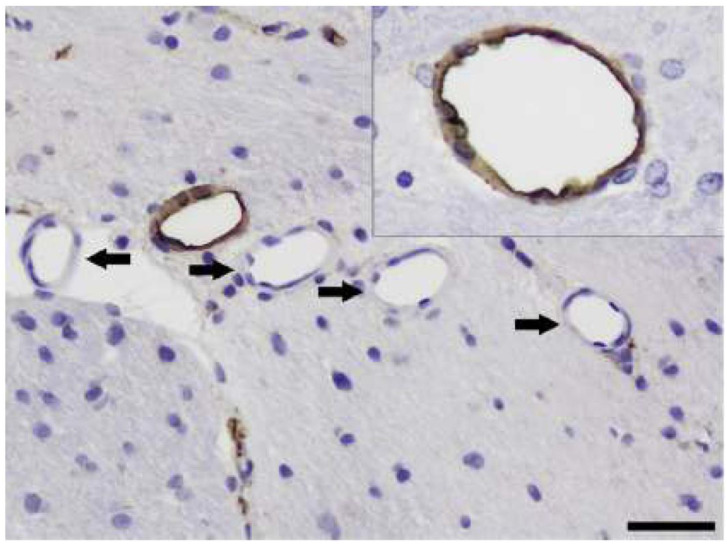
In an ETX-treated rat brain, one microvessel shows normal EBA immunopositivity (higher power view in the inset), but many injured vessels have complete loss of EBA immunolabeling (arrows), confirming ETX-induced vascular damage as EBA is a marker of an intact BRB in rats. EBA immunohistochemistry. Bar = 160 µm.

**Figure 6 vetsci-11-00002-f006:**
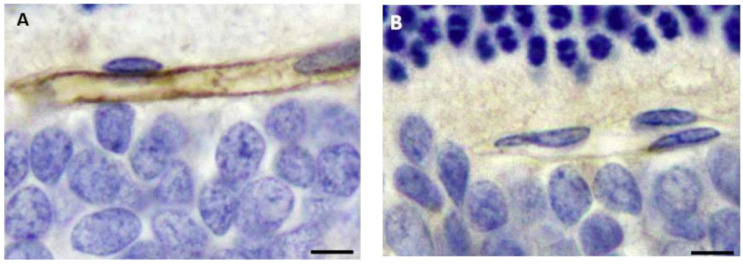
In a control, ETX-untreated rat retina (**A**), an uninjured microvessel shows strong, uniform endothelial EBA immunopositivity while, in an ETX treated retina (**B**), there is marked loss of EBA immunoreactivity in this damaged capillary. EBA immunohistochemistry. Bar = 10 µm. (All of these images are reproduced with permission from articles published in J. Vet. Diagn. Invest. 2014; 26: 470–472 (Figure 1 and Figure 3) [23]; Int. J. Mol. Sci. 2022; 23: 9050 (Figure 2 and Figure 4) [3]; and J. Comp. Pathol. 2018; 158: 51–55 (Figure 5 and Figure 6) [27]. Dr Finnie was an author on all of these publications).

**Table 1 vetsci-11-00002-t001:** *Clostridium perfringens* toxin-based typing classification.

Type	Toxins Produced					
	Alpha	Beta	Epsilon	Iota	Enterotoxin	Necrotic Enteritis B-like
A	+	−	−	−	−	−
B	+	+	+	−	−	−
C	+	+	−	−	+/−	−
D	+	−	+	−	+/−	−
E	+	−	−	+	+/−	−
F	+	−	−	−	+	−
G	+	−	−	−	−	+

**Table 2 vetsci-11-00002-t002:** Retinal layers.

INTERNAL LIMITING MEMBRANE (ILM)
RETINAL NERVE FIBER LAYER
GANGLION CELL LAYER
INNER PLEXIFORM LAYER
INNER NUCLEAR LAYER
OUTER PLEXIFORM LAYER
OUTER NUCLEAR LAYER
EXTERNAL LIMITING MEMBRANE (ELM)
PHOTORECEPTOR LAYERS
RETINAL PIGMENT EPITHELIUM (RPE)
BRUCH’S MEMBRANE
CHORIOCAPILLARIS

## Data Availability

The contents of this review are available on request from the corresponding author.

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
