# Peer review of "Clostridium perfringens Type D Epsilon Toxin Causes Blood–Retinal Barrier Microvascular Damage and Diffuse Retinal Vasogenic Oedema"

_vetsci, 2023, doi:10.3390/vetsci11010002_

Round 1

Reviewer 1 Report

Comments and Suggestions for Authors This is a very well written and straightforward manuscript that will be very helpful for the scientific community. My only comment is that since this is a review, perhaps the author could indicate so at the end of the introduction. Otherwise, in my view, the paper is ready to be published.

Author Response

Thank you. I have indicated in the Introduction that this is a review.

Reviewer 2 Report

Comments and Suggestions for Authors

Introduction: 

Lack of Clarity in Structure: The introduction lacks a clear structure, making it challenging for readers to follow the logical flow of information. Suggestion: Organize the content into subsections, such as "Classification of Clostridium perfringens Toxinotypes," "Epsilon Toxin (ETX): Structure and Activation," and "Pathogenesis of ETX-Induced Diseases," to improve readability and logical progression.

Insufficient Explanation of Key Concepts: The description of epsilon toxin (ETX) and its activation process could be more detailed, lacking a clear explanation of the importance of ETX as the main virulence factor. Suggestion: Provide a more detailed explanation of ETX, emphasizing its significance as the main virulence factor and its specific role in causing neurologic disease. This will enhance the reader's understanding of the toxin's importance.

Incomplete Information on ETX Regulation: The statement about the incomplete understanding of the regulation of ETX production is mentioned without further details. Suggestion: Elaborate on the incomplete understanding of ETX production regulation. Discuss current knowledge gaps and potential implications for understanding the pathogenesis of diseases caused by ETX. Providing context for these gaps will engage the reader.

Limited Discussion on ETX Distribution: The abstract mentions the distribution of circulating toxin to various tissues, including the eye, but lacks specifics on how and why ETX reaches these tissues. Suggestion: Provide more information on the mechanisms involved in the systemic distribution of ETX, especially its targeting of specific organs. Clarify the pathways and factors that contribute to the distribution of ETX throughout the body.

Transition to the Main Focus: The transition from discussing ETX-induced neurologic disease to focusing on ocular injuries is abrupt. Suggestion: Add a transitional sentence to smoothly guide the reader from one aspect to the other. This will improve the coherence of the introduction and help the reader follow the shift in focus.

Absence of Citations in the Introduction: The introduction lacks citations for specific statements, making it challenging for readers to verify the information. Suggestion: Integrate appropriate references to support key points and strengthen the academic rigor of the introduction. Citations will provide credibility and allow readers to explore the referenced literature for further information.

The Blood Retinal- Barrier:

Lack of Specific Examples or Citations: The text lacks specific examples or citations to support statements about the high metabolic activity of the retina, its oxygen consumption, and its ranking in terms of energy consumption among tissues. Suggestion: Integrate specific examples or references to studies that provide evidence for the claims made about the metabolic activity and energy consumption of the retina. This will enhance the credibility of the information presented.

Complex Terminology Without Clarification: The introduction contains complex terminology such as "leaky-type intercellular tight junctions" and "zonulae occludentes" without providing sufficient clarification, potentially hindering understanding for readers unfamiliar with these terms. Suggestion: Include brief explanations or definitions of complex terminology to enhance reader comprehension, ensuring that even those less familiar with the subject matter can follow the content.

Limited Discussion on the Practical Implications: While the text provides detailed information about the blood-ocular barrier, it lacks discussion on the practical implications of this barrier for maintaining retinal health and function. Suggestion: Include a brief discussion on the practical implications of the blood-ocular barrier, such as how it contributes to the protection of the retina from blood-borne toxicants and maintains the optimal environment for visual function.

Insufficient Transition to the Main Topic: The introduction transitions abruptly from discussing the metabolic activity of the retina to describing the components of the blood-ocular barrier. Suggestion: Add a transitional sentence to smoothly connect the discussion on the metabolic activity of the retina with the subsequent information about the blood-ocular barrier. This will improve the overall coherence of this section.

Absence of Recent Citations: The references cited in the text are not explicitly mentioned, and there is a lack of recent citations. Including recent references would ensure that the information is up-to-date and relevant to current scientific understanding. Suggestion: Explicitly mention the key references within the text to provide transparency and allow readers to access the cited literature. Additionally, consider incorporating more recent studies to support the information presented.

Mechanisms of retinal edema formation and clearance:

Complex Terminology Without Clarification: The text introduces complex anatomical terms and processes related to retinal hydration, intraocular pressure, and retinal barriers without providing sufficient clarification for readers unfamiliar with these terms. Suggestion: Include brief explanations or definitions for complex anatomical and physiological terms to enhance reader comprehension. This is particularly important when discussing processes such as transcellular and paracellular transport.

Lack of Specific Examples or Citations: The text lacks specific examples or citations to support statements about the mechanisms of fluid movement within the retina, the role of retinal barriers, and the consequences of altered permeability. Suggestion: Integrate specific examples or references to studies that provide evidence for the mechanisms described, the role of retinal barriers, and the impact of altered permeability. This will strengthen the credibility of the information presented.

Insufficient Discussion on Clinical Relevance: While the text mentions retinal edema as a hallmark feature of ocular diseases, there is limited discussion on the clinical relevance of these processes in the context of human and veterinary eye health. Suggestion: Expand on the clinical implications of retinal edema, providing examples of ocular diseases where these processes play a significant role. This will enhance the practical understanding of the described physiological mechanisms.

Transition Issues Between Paragraphs: The transition between paragraphs discussing retinal water content and mechanisms leading to increased blood-retinal barrier (BRB) permeability is not seamless, making it challenging for readers to follow the logical progression. Suggestion: Add transitional sentences to smoothly guide the reader from one topic to the next. This will improve the overall coherence of the text and help readers navigate the complex information presented.

Limited Discussion on RPE and Muller Cells: Although the text mentions the role of the retinal pigment epithelium (RPE) and Muller cells in fluid resorption, there is limited information about their specific functions and contributions to maintaining retinal hydration. Suggestion: Provide more detailed information about the functions of RPE and Muller cells in fluid resorption. Specify how these cells contribute to the overall regulation of retinal hydration

Epsilon ophthalmotoxicity:

Limited Explanation of Figures: The text references several figures (e.g., Figure 1, Figure 2) without providing adequate explanations or interpretations of the images. This omission may hinder the reader's understanding of the presented electron micrographs and immunohistochemical results. Suggestion: Include concise captions or explanations for each figure, briefly summarizing the key findings and their relevance to the discussed topic. This will enhance the reader's comprehension of the visual data.

Insufficient Clarity in Linking Results: The transition between describing retinal microvascular injury in rats to the heterogeneity of endothelial barrier antigen (EBA) loss lacks clarity. Readers may find it challenging to understand the logical progression from one set of results to the next. Suggestion: Introduce clearer transitional sentences or paragraphs to establish a seamless flow between different sets of experimental results. Provide a brief overview of the main findings before delving into specific details.

Lack of Discussion on Experimental Methods: The text does not provide information on the specific experimental methods used to induce ETX exposure in rats, which is crucial for understanding the validity and reproducibility of the findings. Suggestion: Briefly describe the experimental methods used, including dosages, administration routes, and any relevant controls. This information is essential for readers to assess the rigor of the experimental design.

Sparse Integration of Citations: While the text makes references to prior studies (e.g., ETX-induced retinal damage in rats [17]), it lacks integration of citations to support the correlation between retinal and brain vasogenic edema or the heterogeneity of EBA loss Suggestion: Integrate citations more explicitly to support the correlations and findings discussed. This will enhance the credibility of the presented data and allow readers to explore the referenced studies for additional context.

Unclear Explanation of EBA's Role: The text mentions that EBA is believed to be an important component of tight junctions, but it does not provide a clear explanation of EBA's role or how its loss may impact barrier integrity.

Suggestion: Elaborate on the role of EBA in the context of tight junctions and the blood-retinal barrier. Clarify whether the loss of EBA is a cause or consequence of altered barrier integrity, providing additional insights into its functional significance.

Conclusion:

Lack of Integration with Previous Sections: The conclusion does not effectively tie back to the specific findings and discussions presented earlier in the paper. It provides an overview of the general impact of ETX on the BRB without connecting it to the detailed experimental results or insights gained. Suggestion: Introduce the conclusion by summarizing key findings from the results section, emphasizing their significance in the context of ETX-induced retinal injury. Create a more seamless transition from the experimental details to the broader implications discussed in the conclusion.

Limited Mention of Diagnostic Implications: The conclusion briefly mentions the potential diagnostic utility of retinal pathology, but it lacks a detailed exploration of how this information could be practically applied or contribute to the field. Suggestion: Expand on the idea of diagnostic utility by discussing potential implications for veterinary or medical diagnosis. Address how identifying retinal pathology may aid in diagnosing ETX exposure and related conditions, providing a more comprehensive understanding of its practical applications.

Need for a Forward-Looking Statement: The conclusion does not provide a forward-looking statement or suggest potential directions for future research based on the current findings. Suggestion: Include a brief section or sentence that outlines potential avenues for future research. This can involve addressing remaining questions, investigating related aspects, or exploring the translational implications of the current findings for therapeutic interventions.

Unclear Relationship with Vision Impairment: While the conclusion mentions the likely impairment of vision due to retinal vasogenic edema, it does not elaborate on the specific mechanisms or clinical consequences. Readers may benefit from a more detailed discussion on the potential impact of retinal pathology on vision. Suggestion: Provide a concise explanation of how retinal vasogenic edema may affect vision, considering the structural and functional aspects of the retina. Discuss potential clinical implications for both human and animal subjects, enhancing the understanding of the observed pathology.

Author Response

I thank the reviewers for their helpful comments and these have been addressed in an extensively rewritten paper as follows:

  • The title has been amended (R2) with “in an experimental rat model” added as suggested to denote that the experimental work reviewed in this manuscript was conducted in this species.
  • As suggested by R1, the tissue distribution of ETX has been more fully described (lines 45-49 and 95-98) and its accumulation in the eye referenced (Tamai et al, 2003).
  • The Introduction has been amended (R1) (lines 50-60) describing the structure of the review and relevance of the difference sub-headings.
  • The classification of Clostridium perfringens toxins has been expanded as suggested (R2), and referenced (Rood et al, 2018), with a table (Table 1) included to better outline the toxin-based typing scheme used for this bacterium (lines 62-66).
  • In subsection 3 (Blood-retinal barrier), the vascular supply to the retina in rats has been more fully explained (R1), and referenced (Bhutto et al, 1995), and the vascular pattern in other species (Ofri, 2017) described (lines 129-134).
  • The diseases in which BRB breakdown, and consequent retina edema, are important in humans and animals (Ofri, 2017) has been explained in greater detail (R1) – lines 143-149.
  • An introductory paragraph (lines 151-155) has been added to subsection 4 (Mechanisms of retinal edema formation and clearance) as suggested (R1) to introduce this section. This section has also been extensively rewritten to hopefully provide a better explanation of retina edema formation and clearance. Additional references have been included (Reichenbach et al, 2007; Scholl et al, 2010).
  • A diagrammatic representation of the different retinal layers (Table 2) has been included as suggested (R1) to hopefully permit the reader to better understand the normal route of fluid movement through the retina from the internal limiting membrane to the choroid, and where there are barriers that facilitate (RPE) and impede (external limiting membrane) this fluid passage after BRB breakdown.
  • In Section 5 (Clostridial epsilon ophthalmotoxicity in the rat), the experimental details of the studies conducted in rats have been included (R1), with additional references (Lawrenson et al, 1995; Ghabriel et al, 2000, 2004; and Battacharjee et al, 2002) added to better explain the role of EBA in the BRB.
  • A new section 6 (Future studies) has been added as suggested (R1) to explain how the ocular studies of ETX in the rat could be useful in better characterizing the naturally-occurring disease in ruminants (i.e. whether ophthalmic lesions, yet to be determined, occur in sheep and goats, particularly as blindness is a common clinical sign (Uzal et al, 2016 added), and how these ocular lesions could be diagnostically useful. The potential usefulness of the ETX rat model for testing pharmacologic interventions to treat retinal edema (which are few at present) has also been highlighted.    

Reviewer 3 Report

Comments and Suggestions for Authors

The title is not clear. To which species does the review refer? For rodents? To the sheep and goats? The title must be reshaped so as to immediately understand that these are animal species and above all that the main experiments refer to rodents.

The transition from experiments to hypotheses on goats and sheep is not clear.

For me the scientific part on goats is lacking.

From line 24 to line 27: in the introduction the classification (A,B,C,D,E,F,G) with respect to the toxins produced must be better explained.

For figure 1, for figure 2, for figure 3, for figure 4 and for figure 5 what are the references? What are the sources of the figures? Are they the authors of the review? does the work cited (15) not have these figures within it?

Author Response

(The authors gave the same response as above.)
